# Quantum Confinement Effect in Amorphous In–Ga–Zn–O Heterojunction Channels for Thin-Film Transistors

**DOI:** 10.3390/ma13081935

**Published:** 2020-04-20

**Authors:** Daichi Koretomo, Shuhei Hamada, Yusaku Magari, Mamoru Furuta

**Affiliations:** 1Engineering Course, Kochi University of Technology, Kami, Kochi 782-8502, Japan; 216007n@gs.kochi-tech.ac.jp (Y.M.); furuta.mamoru@kochi-tech.ac.jp (M.F.); 2Material Science and Engineering Course, Kochi University of Technology, Kami, Kochi 782-8502, Japan; 225109c@gs.kochi-tech.ac.jp; 3Center for Nanotechnology, Research Institute, Kochi University of Technology, Kami, Kochi 782-8502, Japan

**Keywords:** oxide semiconductor, thin-film transistor, heterojunction, amorphous, device simulation, quantum confinement

## Abstract

Electrical and carrier transport properties in In–Ga–Zn–O thin-film transistors (IGZO TFTs) with a heterojunction channel were investigated. For the heterojunction IGZO channel, a high-In composition IGZO layer (IGZO-high-In) was deposited on a typical compositions IGZO layer (IGZO-111). From the optical properties and photoelectron yield spectroscopy measurements, the heterojunction channel was expected to have the type–II energy band diagram which possesses a conduction band offset (Δ*E*_c_) of ~0.4 eV. A depth profile of background charge density indicated that a steep Δ*E*_c_ is formed even in the amorphous IGZO heterojunction interface deposited by sputtering. A field effect mobility (*μ*_FE_) of bottom gate structured IGZO TFTs with the heterojunction channel (hetero-IGZO TFTs) improved to ~20 cm^2^ V^−1^ s^−1^, although a channel/gate insulator interface was formed by an IGZO−111 (*μ*_FE_ = ~12 cm^2^ V^−1^ s^−1^). Device simulation analysis revealed that the improvement of *μ*_FE_ in the hetero-IGZO TFTs was originated by a quantum confinement effect for electrons at the heterojunction interface owing to a formation of steep Δ*E*_c_. Thus, we believe that heterojunction IGZO channel is an effective method to improve electrical properties of the TFTs.

## 1. Introduction

Thin-film transistors (TFTs) based on oxide semiconductors (OSs) have attracted considerable attention for next generation flat-panel displays (FPDs) due to their advantages such as high field effect mobility (*μ*_FE_), steep subthreshold swing, and low leakage current [1,2,3,4,5,6,7,8,9]. Although *μ*_FE_ of the OS TFTs is more than an order of magnitude higher than that of hydrogenated amorphous silicon (a-Si:H) TFTs, further improvement of the *μ*_FE_ has been required for OS TFTs to expand their applications [9,10,11,12,13]. Optimization of compositions of the OSs is an approach to improve *μ*_FE_, such as an increase of In ratio in the OSs [9,12,13,14,15]. However, it was reported the trade-off between *μ*_FE_ and the reliability of OS TFTs, because an increase of In ratio in the OSs leads to generation defects such as oxygen vacancies [16,17]. As another approach to improve *μ*_FE_ of the OSs, Koike et al. demonstrated a crystalline ZnO/ZnMgO heterojunction structure that was confirmed to be an improvement of a Hall mobility [18]. Taniguchi et al. reported a heterojunction channel consisting of polycrystalline In–Sn–O on amorphous In–Ga–Zn–O (IGZO) with a type-II energy band lineup, which attributed an improvement of the *μ*_FE_ to ~20 cm^2^ V^−1^ s^−1^ for the TFTs [19].

On the other hand, amorphous OSs have an advantage for the spatial uniformity of their transfer characteristics over a large area; thus, the amorphous OS TFTs are promising candidates for large displays. Various types of heterojunction channels, using amorphous OSs such as IGZO/In–Zn–O (IZO), Hf–In–Zn–O/IZO, Zn–Sn–O/IZO and Al–In–Zn–Sn–O/IZO, have been proposed for TFTs [20,21,22,23,24]. These heterojunction TFTs exhibited excellent *μ*_FE_ of over 30 cm^2^ V^−1^ s^−1^. The *μ*_FE_ improvements were mainly induced by the IZO layer, with high-*μ*_FE_ formed at the channel/gate insulator (GI) interface. Moreover, the heterojunction channels such as IGZO/In–Zn–O/IGZO and In–Ga–Si–O/IGZO/In–Ga–Si–O have also been reported to improve *μ*_FE_ [25,26]. Thus, the heterojunction channels are considered to be an appropriate approach to boost TFT performances even for amorphous OSs. However, the carrier transport properties in the heterojunction channel have not been discussed in detail, despite a key for the improvement of the *μ*_FE_ on the amorphous OS TFTs.

We previously reported about the IGZO TFT with a heterojunction channel consisting of different compositions of the amorphous IGZO films. Based on experimental and simulation results obtained by varying a thickness of the barrier layer of the heterojunction channel, it was found that an IGZO heterojunction channel is an effective method to independently improve electrical properties and the reliability of the TFTs [27]. However, the interface properties of the IGZO heterojunction have not been clearly understood. Furthermore, the influence of a conduction band offset (Δ*E*_c_) at the heterojunction interface on the electrical and carrier transport properties has not been analyzed in detail. Thus, it is worth to understand the steepness and Δ*E*_c_ at the amorphous IGZO heterojunction interface formed by sputtering.

In this research, the electrical properties of the heterojunction IGZO TFTs were investigated by varying the thickness of a high-In IGZO (IGZO-high-In) well layer, which was deposited on a typical composition of the IGZO (IGZO-111) barrier layer for the heterojunction. In addition, formation of the heterojunction interface was considered from both of the experimental and theoretical results. *μ*_FE_ of the IGZO TFTs clearly increased, especially in the low gate voltage (*V*_GS_) region when the heterojunction channel was formed. From a depth analysis of the background charge density, the transition width at the heterojunction interface was estimated to be less than 3 nm. Device simulation analysis revealed that the Δ*E*_c_, which acts as a potential barrier for electrons, strongly affects carrier transport paths in the heterojunction channel, which leads to an improvement in the *μ*_FE_ of the TFTs.

## 2. Experimental Methods

Figure 1a shows a schematic cross-sectional view of the bottom gate structured IGZO TFTs. The IGZO TFTs were fabricated on a heavily doped n-type Si substrate with a 100-nm-thick thermally grown SiO_2_. The doped n-type Si substrate and the SiO_2_ were used as gate electrode and GI, respectively. The IGZO channels were deposited by radio frequency (RF) magnetron sputtering without intentional substrate heating. As shown in Figure 1b,c, homogeneous channel layers of the 10-nm-thick IGZO-111 (homo-IGZO-111) and the 10-nm-thick IGZO-high-In (homo-IGZO-high-In) were separately prepared for comparison with the heterojunction channel. The oxygen flow ratio (O_2_/Ar + O_2_) during the depositions were set at 2% and 49% for the IGZO-111 and the IGZO-high-In channels, respectively. Deposition pressure and RF power for both the channels were maintained at 0.5 Pa and 4.4 W cm^−2^, respectively. For the heterojunction channels as shown in Figure 1d, the IGZO-high-In layer was deposited on the 10-nm-thick-IGZO-111 layer (hetero-IGZO) in a chamber without breaking the vacuum. An upper channel thickness of the IGZO-high-In layer was varied from 2.5 nm to 20 nm, while bottom channel thickness of the IGZO-111 layer was maintained at 10 nm to apply constant electric field at the heterojunction interface. Note here that the deposition conditions for each layer in the heterojunction channel were the same as the homogeneous channels. After formation of the IGZO channel using a shadow mask, a SiO_2_ passivation layer (200 nm) was grown by plasma-enhanced chemical vapor deposition at 180 °C using tetraethoxysilane and O_2_ as precursors. After opening contact holes by photolithography and dry etching, In–Sn–O source/drain electrodes were deposited by sputtering through a shadow mask. Finally, the IGZO TFTs were annealed at 350 °C in ambient air for one hour. Channel width and length were 1000 μm and 690 μm, respectively (W/L = 1000/690 μm).

## 3. Results and Discussion

### 3.1. Crystallinity of IGZO films

First, the crystallinity of each IGZO layer was evaluated by grazing incidence X-ray diffraction (GIXRD, Rigaku Corp., ATX-G, Tokyo, Japan) using Cu-Kα radiation with the X-ray incident beam angle (ω) of 0.35°. Figure 2 shows GIXRD patterns of the (a) IGZO-111 and (b) IGZO-high-In layers as a function of annealing temperature. Both the IGZO-111 and high-In layer showed crystallization at the annealing temperature of 700–800 °C. The XRD peaks obtained from IGZO-111 after 800 °C attributed to crystalline IGZO, whereas that of IGZO-high-In layer mainly related to crystalline In_2_O_3_ due to high In ratio in the IGZO [28,29]. On the other hand, both the IGZO-111 and the IGZO-high-In layers retained their amorphous phase at annealing temperatures lower than 600 °C.

### 3.2. Band Alignment and Steepness at Heterojunction Interface

Optical band gap (*E*_g_) was estimated by Tauc plot. Valence band maximum (VBM) of the IGZO films was estimated from an ionization potential (*I*_p_), which was measured by photoelectron yield spectroscopy (Bunkoukeiki, BIP-KV202GD, Tokyo, Japan). Figure 3a,b show the Tauc plots and photoemission yield of the IGZO layers. The optical band gaps of the IGZO-111 and the IGZO-high-In layers were estimated to be ~3.1 and ~2.8 eV, respectively. VBMs of the IGZO-111 and the IGZO-high-In layers were ~7.4 and ~7.5 eV, respectively. Since conduction band minimum (CBM) of IGZO mainly consists of In-5s orbitals [30,31,32], CBM of the IGZO layer would be influence by an In ratio. From these results, the energy band diagram of the IGZO-111 and the IGZO-high-In layers was shown in Figure 3c, suggesting that a Δ*E*_c_ of ~0.4 eV might be formed at the heterojunction interface.

To confirm steepness at the heterojunction interface, the depth profile of background charge density (*N*_bg_) was evaluated by a Schottky diode (SD) with the heterojunction channel as shown in Figure 4a. Detailed fabrication process of the IGZO SD was reported elsewhere [33]. For the SD with the heterojunction channel, channel thicknesses for the each IGZO layer were set at 30 nm. Carrier density (*n*_e_) of the IGZO-111 was set at ~1 × 10^17^ cm^−3^ to obtain good Schottky contact, while that of the IGZO-high-In was set at ~3 × 10^18^ cm^−3^ to enhance the difference of *N*_bg_ at the heterojunction interface. The *N*_bg_ was estimated using an equation as shown below [34,35];
(1)$$N_{bg}=\frac{2}{q\varepsilon_{0}\varepsilon_{s}}\left[\frac{-1}{\partial \left(1/C_{s}^{2}\right)/\partial V}\right]$$
where *ε*_s_ and *C*_s_ were permittivity and capacitance per unit area of the IGZO, respectively. The *C*_s_ was measured using Agilent E4980A LCR meter (California, CA, USA). Figure 4a shows 1/*C_s_*^2^-*V* characteristic of the Schottky diode at 1 kHz. The 1/*C_s_*^2^ values gradually decreased when the voltage applied from −1.5 to 0 V. This result indicates that the heterojunction channel was not fully depleted at the negative voltage of −1.5 V. As the voltage exceeded 0 V, the 1/*C_s_*^2^ values decreased steeply.

From the 1/*C_s_*^2^-*V* characteristic, depth profile of *N*_bg_ in the heterojunction channel was calculated as shown in Figure 4b. The *N*_bg_ of IGZO-111 layer at the depletion depth from 10 to 30 nm showed approximately 1 × 10^17^ cm^−3^, and the *N*_bg_ of IGZO-high-In steeply increased to ~2 × 10^18^ cm^−3^ when the depletion depth exceeds 30 nm. The region where the transition of *N*_bg_ occurred corresponded to the heterojunction interface region. Moreover, the transition width of *N*_bg_ at the heterojunction interface was estimated to be less than 3 nm. This result indicated that steep Δ*E*_c_ would be formed at the heterojunction interface even though both the amorphous IGZO layers were deposited by sputtering.

### 3.3. TFT Characteristics

Electrical properties of the homogeneous IGZO (homo-IGZO) TFTs were first examined as references for comparison with the heterojunction IGZO (hetero-IGZO) TFTs. Transfer characteristics of the TFTs were measured in dark using precision semiconductor parameter analyzers (Agilent 4156C and 4156A, California, CA, USA). The threshold voltage (*V*_th_) was defined as the *V*_GS_ required to obtain the drain current (*I*_DS_) of 1 nA in a linear region. The hysteresis (*V*_H_) was extracted from *V*_th_ difference between forward and reverse characteristics. The *μ*_FE_ was calculated from a linear region using the following equation;
(2)$$\mu_{FE}=\frac{g_{m}}{\frac{W}{L}C_{i}V_{DS}}$$
where *g*_m_, *C*_i_, and *V*_DS_ denote transconductance, capacitance of GI per unit area, and drain voltage, respectively.

Figure 5 shows the transfer characteristics of the homo-IGZO-111 and the IGZO-high-In TFTs. Owing to the optimization of a channel thickness and an oxygen flow ratio during the channel deposition [36,37], *V*_th_ of the homo-IGZO-111 and the IGZO-high-In TFTs were close to zero. From *μ*_FE_-*V*_GS_ curve shown in Figure 5, *μ*_FE_ of both the homo-IGZO TFTs increased with increasing *V*_GS_. The homo-IGZO-high-In TFT exhibited *μ*_FE_ of 22.9 cm^2^ V^−1^ s^−1^, which is approximately two times higher *μ*_FE_ than that of the homo-IGZO-111 TFT (12.4 cm^2^ V^−1^ s^−1^). This result indicates that *μ*_FE_ improvement of the homo-IGZO-high-In TFT was originated by a high-In composition, since large spherical In 5s orbitals mainly form carrier transport paths [26,30,31].

Next, electrical properties of the hetero-IGZO TFTs consisting of the IGZO-high-In on the IGZO-111 were explored. Figure 6 shows the transfer characteristic and *μ*_FE_-*V*_GS_ curves of the hetero-IGZO TFTs with various IGZO-high-In thicknesses. The electrical properties of the hetero-IGZO TFTs summarized in Table 1. The hetero-IGZO TFT with a 2.5-nm-thick upper IGZO-high-In layer showed a *μ*_FE_ of 9.9 cm^2^ V^−1^ s^−1^, which is almost the same *μ*_FE_ as homo-IGZO-111 TFT. In contrast, *μ*_FE_ of the hetero-IGZO TFT significantly improved to 17.2 cm^2^ V^−1^ s^−1^ when the upper IGZO-high-In thickness increased from 2.5 to 5.0 nm. The hetero-IGZO TFT with a 10-nm-thick upper IGZO-high-In layer exhibited a *μ*_FE_ of 19.6 cm^2^ V^−1^ s^−1^. When the upper IGZO-high-In thickness further increased to more than 10 nm, the *μ*_FE_ saturated to about 21 cm^2^ V^−1^ s^−1^, while *V*_th_ started to shift in the negative *V*_GS_ direction. It can be considered that higher negative *V*_GS_ is required to deplete the channel when the upper IGZO-high-In thickness increase, because the carrier density of the IGZO-high-In layer is higher than that of the IGZO-111 layer. Note here that the *μ*_FE_ mainly improved in low-*V*_GS_ region (*V*_GS_ ≦ 10 V), and it decreased to a similar value as the homo-IGZO-111 TFT at *V*_GS_ of 20 V. As a result, the *μ*_FE_-*V*_GS_ curves showed a peak at *V*_GS_ below 10 V. The *μ*_FE_ improvement would be caused by carrier transport in the high-In layer owing to a formation of steep Δ*E*_c_ at the heterojunction interface. To understand an origin of the *μ*_FE_ improvement in the hetero-IGZO TFTs, carrier transport in the heterojunction channel is discussed based on the results obtained using a device simulation (Atlas, Silvaco Inc., California, CA, USA).

Meanwhile, there is no *V*_H_ in the hetero-IGZO TFTs, although the homo-IGZO-high-In TFT showed *V*_H_ of +1.2 V as shown in Figure 5. These results indicate that the hetero-TFTs showed high *μ*_FE_ with an improvement of reliability. The reliability results of the IGZO TFTs are discussed in the Appendix A.

### 3.4. Device Simulation

To understand carrier transport in the hetero-IGZO TFTs, device simulation was carried out using device simulator, Atlas. For the model of IGZO TFTs, a *n*_e_-dependent mobility (*μ*_d_) model with the density of states (DOSs) model was proposed by K. Abe [36]. The *n*_e_-dependent *μ*_d_ model is given by the following equations [36];
(3)$$\mu_{d}=\mu_{d0}\left(1+\gamma \right)\left(\frac{n_{e}}{n_{CR}}\right)$$
(4)$$\gamma =\frac{T_{\gamma }}{T}+\gamma_{0}$$
where *μ*_d0_ is the intrinsic mobility and *n*_CR_ is the critical carrier density, and γ is the index parameter.

The simulation parameters, namely, band gap, electron affinity, and *n*_e_ dependences of *μ*_d_ were defined by the experimental results for both the IGZO layers (see Appendix A). The transfer characteristics of the homo-IGZO TFTs shown in Figure 5 were first reproduced to determine the DOSs in each IGZO layer [38,39,40]. The extracted parameters in both the IGZO models are shown in Table 2. Then, transfer characteristic of the hetero-IGZO TFT with a 10-nm-thick upper high-In layer was reproduced using the same DOSs as homo-IGZO TFTs. Note here that the contacts between S/D electrodes and IGZO back channel were assumed to be ohmic [41].

From above mentioned experimental results, the cause of the *μ*_FE_ improvement in the hetero-IGZO TFTs is considered to be related to a formation of Δ*E*_c_ at the heterojunction interface. To prove the hypothetical origin of the *μ*_FE_ improvement, influence of the Δ*E*_c_ on carrier transport paths of the hetero-IGZO TFT with a 10-nm-thick upper IGZO-high-In layer was analyzed by a device simulation. Figure 7a,b show transfer characteristics and *μ*_FE_-*V*_GS_ curves of the hetero-IGZO TFTs as a function of Δ*E*_c_ formed at the heterojunction interface. The drain current and *μ*_FE_ of the hetero-IGZO TFTs increased with increasing the Δ*E*_c_. When the Δ*E*_c_ was set at zero eV, the *μ*_FE_ of the hetero-IGZO TFT gradually increased with increasing *V*_GS_ and showed approximately 10 cm^2^ V^−1^ s^−1^ at the *V*_GS_ of 20 V. In contrast, the *μ*_FE_ in low-*V*_GS_ region improved when the value of Δ*E*_c_ increased. Moreover, the *V*_GS_ value obtained at maximum *μ*_FE_ shifted to a positive *V*_GS_ direction when the Δ*E*_c_ increased from 0.1 to 0.45 eV. The experimental *μ*_FE_-*V*_GS_ curve could be reproduced well when the Δ*E*_c_ was set at 0.45 eV. Thus, it was confirmed that the Δ*E*_c_ at the heterojunction interface strongly influenced *μ*_FE_ of the hetero-IGZO TFT especially in low-*V*_GS_ region.

To clarify influence of the Δ*E*_c_ on carrier transport properties, drain current densities in the hetero-IGZO TFTs with different Δ*E*_c_ were also analyzed by a device simulation as shown in Figure 8. The drain current densities in the hetero-IGZO TFTs were extracted at the applied *V*_GS_ of 10 V and *V*_DS_ of 0.1 V. For the hetero-IGZO TFT with a Δ*E*_c_ of zero eV, the drain current densities increased at the IGZO−111/GI interface. Therefore, the hetero-IGZO TFT without Δ*E*_c_ exhibited almost the same *μ*_FE_ as homo-IGZO-111 TFT. On the other hand, the drain current densities at the IGZO-111/GI interface reduced and that at the heterojunction interface increased with increasing the Δ*E*_c_. These results indicate that the carrier transport in the hetero-IGZO-TFTs changed from the IGZO-111/GI interface to the heterojunction interface owing to a quantum confinement effect for electrons when the Δ*E*_c_ was formed at the heterojunction interface. Thus, the experimental and device simulation results clarified that the improvement of *μ*_FE_ in the hetero-IGZO TFTs was mainly caused by a quantum confinement effect for electrons, which was induced by Δ*E*_c_ at the heterojunction interface.

## 4. Conclusions

We investigated the electrical properties of the TFTs with the IGZO heterojunction channels consisting of different compositions. The heterojunction channel was formed to deposit a high-In composition IGZO layer on an IGZO-111 layer. From band alignment analyses, type-II energy band diagram was expected to form at the heterojunction interface with a Δ*E*_c_ of ~0.4 eV. In addition, a depth profile of background charge density indicated that a steep Δ*E*_c_ was formed at the amorphous IGZO heterojunction interface formed by sputtering. The *μ*_FE_ of the IGZO TFT with the heterojunction channel improved to 20.1 cm^2^ V^−1^ s^−1^, and its *μ*_FE_-*V*_GS_ curve exhibited a maximum value at applying *V*_GS_ of ~10 V. The experimental *μ*_FE_-*V*_GS_ curve could be reproduced well by a device simulation when the Δ*E*_c_ was assumed at the heterojunction interface. Moreover, the device simulation results indicated that the carrier transport in the hetero-IGZO-TFTs changed from the IGZO-111/GI interface to the heterojunction interface owing to a quantum confinement for electrons when the Δ*E*_c_ was formed at the heterojunction interface. Thus, we believe that heterojunction IGZO channel provides an effective method to improve electrical properties of the TFTs.

## Figures and Tables

**Figure 1 materials-13-01935-f001:**
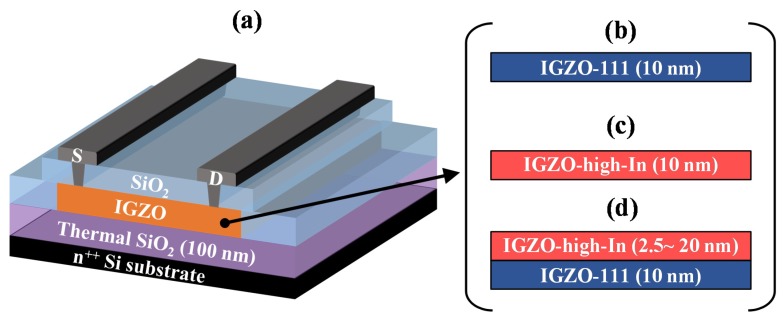
(**a**) Schematic cross-sectional view of the bottom gate structured In–Ga–Zn–O thin-film transistor (IGZO TFT). Channel structures of the (**b**) homogenous typical composition of the IGZO (homo-IGZO-111), (**c**) homogenous high-In IGZO (homo-IGZO-high-In), and (**d**) heterojunction IGZO (hetero-IGZO).

**Figure 2 materials-13-01935-f002:**
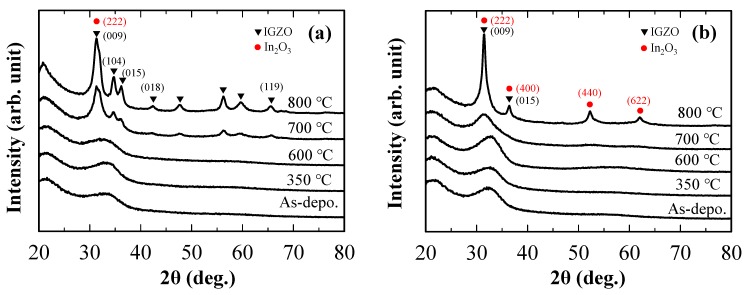
Grazing incidence X-ray diffraction (GIXRD) patterns of (**a**) IGZO-111 and (**b**) IGZO-high-In layers as a function of annealing temperature. Annealing was carried out in air for one hour.

**Figure 3 materials-13-01935-f003:**
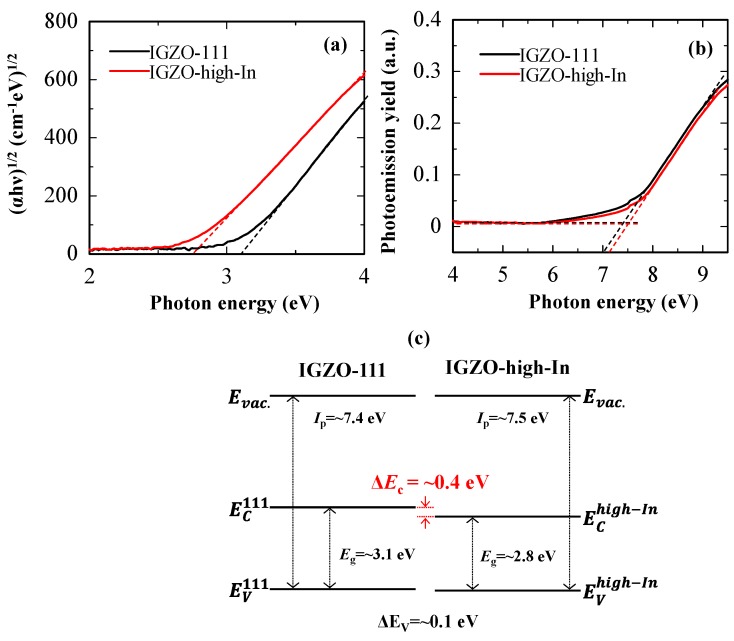
(**a**) Tauc plots and (**b**) photoemission yield of the IGZO-111 and the IGZO-high-In layers after annealing at 350 °C in in ambient air for one hour. (**c**) Energy band diagrams of the IGZO-111 and the IGZO-high-In layers.

**Figure 4 materials-13-01935-f004:**
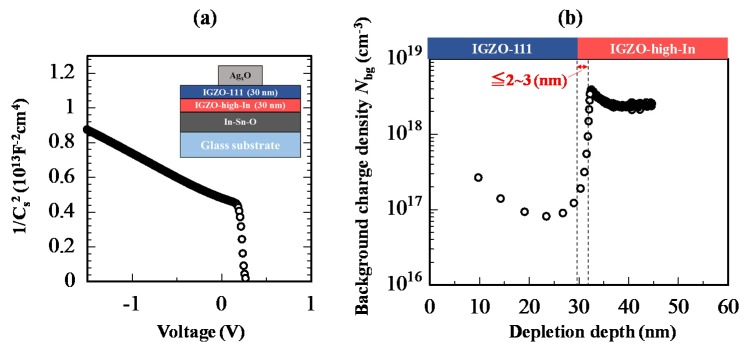
(**a**) 1/*C_s_*^2^-*V* characteristic of the Schottky diode at 1 kHz. Inset is a schematic cross-sectional view of the Schottky diode with the heterojunction IGZO. The heterojunction IGZO was annealed at 350 °C in ambient air for one hour before deposition of an Ag*_x_*O electrode. (**b**) Depth profile of *N*_bg_ calculated using the 1/*C_s_*^2^-*V* characteristic. The depth *x* = 0 nm corresponds to the Ag*_x_*O/IGZO-111 interface.

**Figure 5 materials-13-01935-f005:**
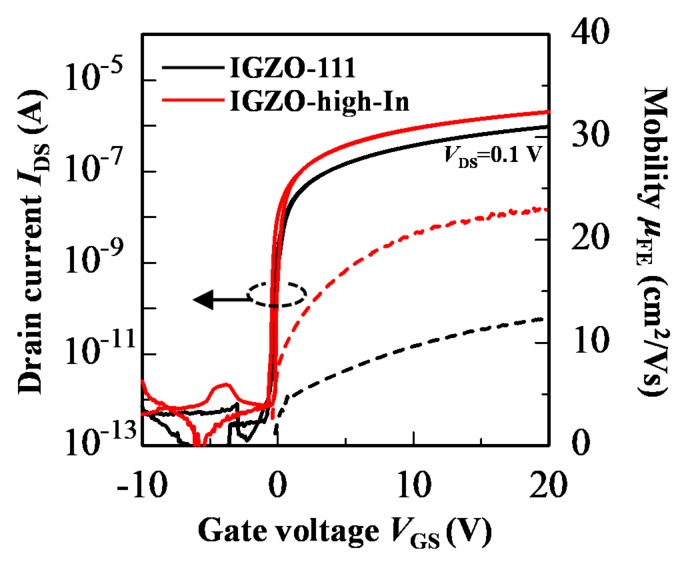
Transfer characteristics of the homo-IGZO-111 and the IGZO-high-In TFTs (*V*_DS_ = 0.1 V, W/L = 1000/690 μm).

**Figure 6 materials-13-01935-f006:**
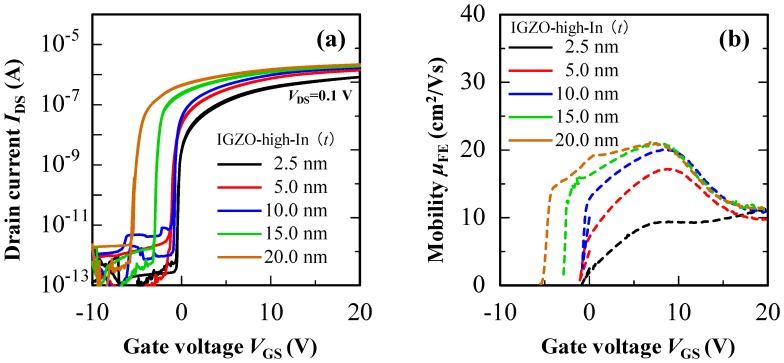
(**a**) Transfer characteristics and (**b**) *μ*_FE_-*V*_GS_ curves of the hetero-IGZO TFTs with various thickness of the IGZO-high-In layer (*V*_DS_ = 0.1 V, W/L = 1000/690 μm).

**Figure 7 materials-13-01935-f007:**
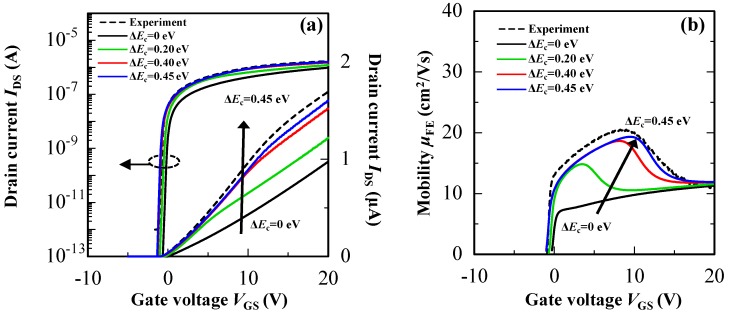
Simulation results of (**a**) transfer characteristics and (**b**) *μ*_FE_-*V*_GS_ curves of the hetero-IGZO TFTs with different Δ*E*_c_. The broken line is an experimental result of the hetero-IGZO TFT.

**Figure 8 materials-13-01935-f008:**
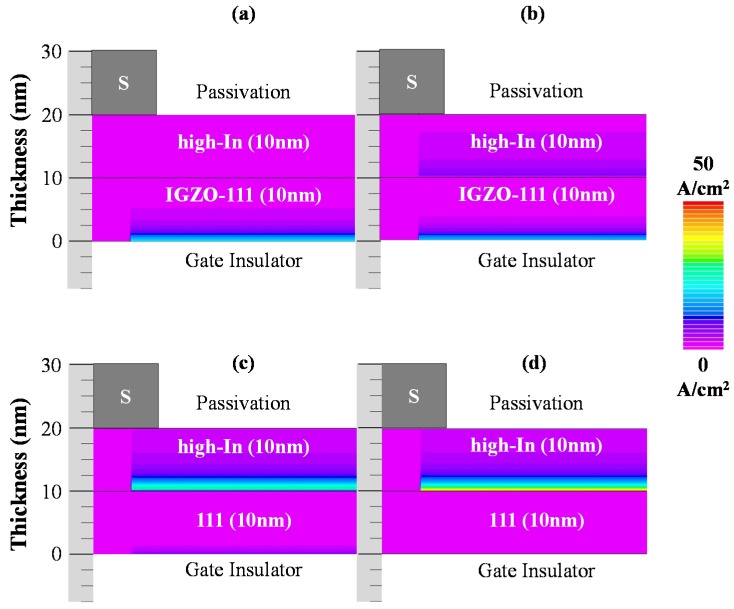
Drain current densities in the hetero-IGZO TFTs with Δ*E*_c_ of the (**a**) 0 eV, (**b**) 0.2 eV, (**c**) 0.4 eV, and (**d**) 0.45 eV. The insert is the color coding for the drain current density at *V*_GS_ = +10 V (*V*_DS_ = 0.1 V).

**Table 1 materials-13-01935-t001:** Summary of electrical properties of the hetero-IGZO TFTs with the various IGZO-high-In thicknesses.

IGZO-High-In Thickness	2.5 nm	5 nm	10 nm	15 nm	20 nm
IGZO-111 Thickness	10 nm				
*μ*_FE_ (cm^2^ V^−1^ s^−1^)	9.9	17.2	19.6	21.8	21.3
*S.S.* (V/dec.)	0.10	0.10	0.10	0.12	0.15
*V*_th_ (V)	0	−0.9	−0.9	−3.3	−5.1
*V*_H_ (V)	0.0	0.0	0.0	0.0	0.0

**Table 2 materials-13-01935-t002:** Extracted parameters in the homo-IGZO-111 and the IGZO-high-In models.

Symbol	Value (IGZO)		Unit	Description
	−111	-High-In		
*N* _C_	5.0 × 10^18^	5.0 × 10^18^	cm^−3^	Effective conduction band density of states
*μ* _d0_	14	30	cm^2^ V^−1^ s^−1^	Intrinsic electron mobility
*n* _CR_	1.0 × 10^20^	1.0 × 10^20^	cm^−3^	Critical electron density
*T* _γ_	178.4	178.4	K	γ temperature
γ_0_	−0.31	−0.31	—	Gamma at 1/T = 0
*W* _ga_	0.7	1.2	eV	Decay energy of acceptor-like Gaussian trap
*W* _gd_	0.12	0.12	eV	Decay energy of donor-like Gaussian trap
*E* _ga_	0	0	eV	Mean energy of Gaussian acceptor-like trap
*E* _gd_	2.6	2.2	eV	Mean energy of Gaussian donor-like trap
*N* _ga_	1.5 × 10^17^	1.5 × 10^17^	cm^−3^ ev^−1^	Peak density of Gaussian acceptor-like trap
*N* _gd_	1.3 × 10^17^	1.3 × 10^17^	cm^−3^ ev^−1^	Peak density of Gaussian donor-like trap
*N* _ta_	1.0 × 10^19^	1.0 × 10^19^	cm^−3^ ev^−1^	Acceptor-like tail trap density
*N* _td_	3.0 × 10^19^	3.0 × 10^19^	cm^−3^ ev^−1^	Donor-like tail trap density
*W* _ta_	0.01	0.01	eV	Slope of acceptor-like tail trap
*W* _td_	0.1	0.1	eV	Slope of donor-like tail trap

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
