# Peer review of "Quantum Confinement Effect in Amorphous In–Ga–Zn–O Heterojunction Channels for Thin-Film Transistors"

_materials, 2020, doi:10.3390/ma13081935_

Round 1

Reviewer 1 Report

The paper is well-organized and the results appear interesting.

My comments and suggestions are outlined below.

1) The originality of this study and its contribution to the field with respect to ref. 27 should be better highlighted.

2) The practical reliability of a TFT performing as in Table 1 need to be better discussed in the text.

3) The modelling analysis based on the use of ATLAS Silvaco tools should be properly presented in Section 3.4.

For example, to recall the key physical models and also the DoS model, cite the following recent papers: 

[x] F. Pezzimenti, “Modeling of the steady state and switching characteristics of a normally-off 4H-SiC trench bipolar-mode FET” IEEE Transactions on Electron Devices, vol. 60, p. 1404-1411, 2013.

[x] M. K. Anvarifard and  A. A. Orouji, “Proper Electrostatic Modulation of Electric Field in a Reliable Nano-SOI With a Developed Channel” IEEE Trans. Electron Dev., 65, 1653, 2018.

[x] H. Bencherif, L. Dehimi, F. Pezzimenti, F. G. Della Corte, “Temperature and SiO2/4H-SiC interface trap effects on the electrical characteristics of low breakdown voltage MOSFETs” Appl. Phys. A-Mater., vol. 125, p. 294, 2019.

4) Fig. 7(a) should be resized.

Reviewer 3 Report

Koretomo et al. present a complete work deling with the optimization of  a INGaZnO thin film transistor. Experimentally, they compare different devices with the same architecture but different active media with the corresponding study of their electronic and optical properties. Finally, the authors use the experimental outputs to make simulations of a range of devices to predict the behaviour of the simulated devices, such as the current density in different conditions of quantum confinement.

I belive, this strategy of getting basic parameters experiment and using simulations for study/desing of devices is very usful in this particular field of microelectronics, and on top of that the authors have provided a very detailed explanations of the process. As a result, I should recommend the manuscript to be published as it is.

Kind Regards

Author Response

We would like to thank you for your time for reviewing our manuscript, and also thank you for positive comment. We believe these findings in “Quantum Confinement Effect in Amorphous In–Ga–Zn–O Heterojunction Channel for Thin-Film Transistor”, will provide valuable information in this field. We tried to improve our paper quality based on another reviewer comments.

Reviewer 4 Report

This paper reports the electrical and carrier transport properties in an In–Ga–Zn–O thin-film transistors (IGZO TFTs) with heterojunction channel. Optimized TFTs exhibit a  field effect mobility of ~20 cm2V-1s-1, significantly higher than that of TFT based on IGZO-111. However, previous paper published by the authors has studied the advantage of heterojunction channel. The mian difference is the change of thickness of high-In IGZO of IGZO-111. Also, introduce of high-In IGZO layer would reduce device stability, e.g NIBS, PIBS, shelf-storage stability. This should be the mian topic for a following work.

  1. The backward scans in transfer curves should be provided.
  2. XPS technique is strongly recommended to determine the WF and valence band maximum
  3. Figure 2. Please label the peaks for each oxide. Why the high-In IGZO exhibit lower crystallinity than IGZO-111 upon below 700 C?
  4. the details how to calculate the depth profile trap density and 1/C2-V should be provided.

Reviewer 5 Report

In this work, devices with different architectures are fabricated and analyzed for TFT applications. Analysis of using IGZO and IGZO-high-In layers with different thicknesses are reported. The authors observe an increase of the field effect mobility for the case of ticker semiconductor layer devices. This increase is attributed to the variation of conductance energy at the interface due to electron confinement.

The work is interesting for the design of TFTs and is well distributed and clearly explained. However some change are required by this reviewer before being published.

The point 2.2 details some experiments and formulas. It is weird to see in this part. This reviewer recommends to describe the mobility equation directly in the point 3.3.

In line 134, the authors calculate the Nbg using 1/C2-V characteristic. Please expand this calculations to facilitate the comprehension. Maybe 1/C2-V could help.

In Figure 5, the resulting field effect mobility seems gradually increase, suggesting an increase of the transconductante. Please include a graph with the transconductance to verify that these mobility increase is not due to the channel resistance reduction which increases according to the back gate voltage. Is the drain current well saturated?

Do this electron confinement reported in other heterostructures with oxide semiconductors?

Have the author check or report any influence of the series resistance in the field effect mobility? The threshold voltage shift can also be important. Do they have different length or width devices?

Round 2

Reviewer 4 Report

As I mentioned previously. The paper published by the authors in Japanese JAP has studied the advantage of heterojunction channel. The mian difference is the change of thickness of high-In IGZO of IGZO-111. The introduce of high-In IGZO layer would reduce device stability, e.g NIBS, PIBS, shelf-storage stability. This should be the mian topic for a following work. The authors did not work on these problems, thus I would perfer rejecting it.

Reviewer 5 Report

Thank you for following the suggestions. For this reviewer the manuscript can be published in the present form. 

Author Response

Thank you very much for your quick and positive review.
we will improve our manuscript based on the comments of other reviewers.

Best regards,

Daichi Koretomo